# Opposite Root Morphological Responses of Chinese Cabbage to Poly-γ-glutamic Acid When Applied with Urea and Ammonium Sulphate

**Lei Zhang [1,2], Lingli Wang [1], Yu Sun [3], Xiaoyan Dong [4], Zhanbo Wei [1], Lili Zhang [1,*] and Yuanliang Shi [1,*]**

1 Institute of Applied Ecology, Chinese Academy of Sciences, 72 Wenhua Road, Shenyang 110016, China
2 Postdoctoral Research Station, Shikefeng Chemical Co., Ltd., 101 Shanjian Road, Linyi 276002, China
3 Key Laboratory of Mollisols Agroecology, Northeast Institute of Geography and Agroecology, Chinese Academy of Sciences, 4888 Shengbei Road, Changchun 130102, China
4 Yantai Institute of Coastal Zone Research, Chinese Academy of Sciences, 17 Chunhui Road, Yantai 264003, China
* Correspondence: llzhang@iae.ac.cn (L.Z.); shiyl@iae.ac.cn (Y.S.); Tel.: +86-139-4006-6843 (L.Z.); +86-137-0402-2984 (Y.S.)

**Abstract:** Poly-γ-glutamic acid (γ-PGA) significantly promotes the fertilizer N uptake efficiency of crops and evidently affects soil available N (nitrate, ammonium and glutamate) status. As an adaptive strategy to forage N, root morphology responds variably to soil available N dynamics. Detailed knowledge of how root morphology responds to γ-PGA remains unexplored. A pot trial was conducted to investigate the response of root morphological traits to γ-PGA when applied with $CO(NH_2)_2$ or $(NH_4)_2SO_4$. The results showed that γ-PGA significantly improved the dry weight, total carbon and total nitrogen content of roots, and with a higher improvement with $CO(NH_2)_2$ compared to $(NH_4)_2SO_4$. γ-PGA significantly increased the root length, total surface area, tips and forks with $CO(NH_2)_2$. Contrarily, γ-PGA significantly reduced root length, specific root length, specific root area, root volume, tips and forks with $(NH_4)_2SO_4$, with its inhibition on root growth mainly caused by the high ammonium content in soil. In conclusion, γ-PGA gives opposite effects on root morphological traits when applied with $CO(NH_2)_2$ or $(NH_4)_2SO_4$. This finding provides a new insight to reveal the promotion mechanism of γ-PGA on plant N acquisition in the rhizosphere and offers a practical reference for optimizing γ-PGA and fertilizer application management.

**Keywords:** poly-γ-glutamic acid (γ-PGA); root morphology; urea; ammonium sulphate; glutamate

## 1. Introduction

For plant growth, nitrogen (N) is one of the most important nutrients, and its availability is a major constraint. Nitrogen is available in many different forms in soil, but the three N forms most utilized by roots are nitrate ($NO_3^-$-N), ammonium ($NH_4^+$-N) and amino acids [1]. Roots are responsible for taking nutrients from soil and have a strong morphological plasticity for changing soil environments, which is an adaptive strategy for resource foraging and also a key factor limiting efficient N acquisition by crops [2,3]. Many studies have revealed complex responses of root growth and development to changing N supplies [4–8]. Soil available inorganic N ($NH_4^+$-N and $NO_3^-$-N) affects the root morphology development in a dose-dependent manner. As reported by Gruber et al. [9], a mild N deficiency causes the primary root and lateral root to grow longer, but a severe N deficiency inhibits the primary root and lateral root to grow longer and the lateral root to emerge. A superfluous supply of $NH_4^+$-N (>5mM) and $NO_3^-$-N (>10 mM) both result in a systemic repression of root growth, where high $NH_4^+$-N and high $NO_3^-$-N repress mainly primary root elongation and lateral root elongation, respectively [8,10,11]. Moreover, $NH_4^+$-N and $NO_3^-$-N appear to stimulate lateral root development in a complementary way, because

the former stimulates branching, while the latter stimulates elongation and density [10]. In addition, different forms of N fertilizer go through different N transformation processes in soil and result in different available N status in soil [12], thus induce different root morphological responses [13,14].

As previously reported, most amino acids have a general inhibition phenomenon on root growth [6], however, only L-glutamic acid (L-Glu) has a unique and highly specific effect on root morphology and root branching [6,15]. Walch-Liu et al. [15] confirmed that L-Glu even at very low concentrations (0.05 mM) was capable of producing a shorter, more branched root system by inhibiting primary root growth and stimulating LR outgrowth. A recent study further demonstrated that L-Glu at the micromolar level (0.01 to 0.1 mM) could arrest PR elongation and total root length in B. napus [8]. Furthermore, some other studies have revealed that soil mineral N and L-Glu have complex and linked effects on root growth and development [11,16]. These distinct responses of root morphological traits on soil available N status reflect that crops have different strategies to deal with various N availability [7].

$\gamma$-PGA is a kind of naturally occurring polymer produced by microorganisms, only composed of Glu and connected by linking the $\alpha$-amino group and the $\gamma$-carbonyl group of two adjacent Glu [17]. After merging with soil, $\gamma$-PGA is degraded by soil microorganisms gradually, subsequently resulting in an explosion of Glu content in soil [18]. As a synergistic fertilizer or as a plant growth regulator, $\gamma$-PGA can significantly increase the N uptake by plants, and thus improve the plant growth and N use efficiency [19]. As reported in our previous studies, $\gamma$-PGA reduced the $NH_4^+$-N content in soil and delayed the release of $NO_3^-$-N by firstly reducing and subsequently increasing the $NO_3^-$-N content in soil [20], and $\gamma$-PGA evidently changed the fate of fertilizer N in plant and soil N pools by changing the N transformation process in soil, clarifying the process of $\gamma$-PGA in improving N uptake efficiency and reducing N loss [19]. In addition, some other previous studies demonstrated that $\gamma$-PGA increased the $NH_4^+$-N and reduced the $NO_3^-$-N content in soil [21,22]. Overall, previous studies show that $\gamma$-PGA has a profound effect on the soil N cycle by changing the content of $NH_4^+$-N, $NO_3^-$-N and microbial N in soil [20–23], and differently altering the N fate to soil N pools when applied with different N fertilizers [19]. Additionally, some earlier studies found that $\gamma$-PGA increased the root biomass and root activity when applied with urea ($CO(NH_2)_2$) [20,23].

Clearly, plants have evolved sophisticated morphological traits to sense and respond to changes in different components of external N supply, and the addition of $\gamma$-PGA could evidently change the $NH_4^+$-N, $NO_3^-$-N and L-Glu contents in soil. Therefore, it can be hypothesized that $\gamma$-PGA would alter the root morphological traits and regulate them differently when used with different forms of N fertilizers. Yet, what effect $\gamma$-PGA exerts on root morphology remains unexplored, especially when applied with different chemical fertilizers. In order to test this hypothesis, we conducted a pot trial to investigate the effects of $\gamma$-PGA on root morphological traits by applying $\gamma$-PGA with $CO(NH_2)_2$ or ammonium sulphate (($NH_4)_2SO_4$). The objective of this study was to elucidate the response of root morphological traits on $\gamma$-PGA addition and to reveal the different responses when it was applied with different N fertilizers.

## 2. Materials and Methods

Four treatments were set up in this incubation experiments: (1) $CO(NH_2)_2$ application alone; (2) $\gamma$-PGA with $CO(NH_2)_2$ application; (3) ($NH_4)_2SO_4$ application alone; (4) $\gamma$-PGA with ($NH_4)_2SO_4$ application. Each treatment dealt with 10 replicates. The soil used in this study is a aquic brown soil and classified as Hapli-Udic Alfisol or Luvisol [24], has a pH 4.98, total carbon (TC) of 12.66 g kg$^{-1}$, total nitrogen (TN) of 1.22 g kg$^{-1}$, $NO_3^-$-N of 24.31 mg kg$^{-1}$, $NH_4^+$-N of 15.40 mg kg$^{-1}$, Olsen P of 32.80 mg kg$^{-1}$, available K of 121.57 mg kg$^{-1}$, clay of 15.80%, silt of 77.00% and sand of 7.20%. Details are as follows. The pot size was 18 cm diameter and 11 cm height, 1 kg soil was filled in each pot and three Pakchoi seeds (*Brassica rapa* subsp. *chinensis*) were sowed in each pot. P and K fertilizers

were applied as basal fertilizer, and 10 days after the basal fertilization, the N fertilizer with or without γ-PGA in solution was applied to the pot after the Pakchoi seedlings (*Brassica rapa* subsp. *chinensis*) were thinned to 1 plant. The γ-PGA, N, P and K fertilizer were applied at the rate of 0.3 g kg$^{-1}$ soil, 0.16 g N kg$^{-1}$ soil, 0.067 g P$_2$O$_5$ kg$^{-1}$ soil and 0.13 g K$_2$O kg$^{-1}$ soil, respectively. The solid γ-PGA was purchased from Xuankai Bio-technology Co., LTD, Nanjing, China, with a purity of 99% and a molecular weight of 1000 kD; the N fertilizers were CO(NH$_2$)$_2$ or (NH$_4$)$_2$SO$_4$, the P fertilizer was Ca(H$_2$PO$_4$)$_2$ and the K fertilizer was K$_2$SO$_4$. To eliminate the possible influence of γ-PGA-N on root growth, an equivalent amount of CO(NH$_2$)$_2$-N or (NH$_4$)$_2$SO$_4$-N was added to soils without γ-PGA addition at the time of N fertilization. All pots were placed in a controlled growth greenhouse with a 12 h photoperiod, a constant humidity of 55% ± 1% and a temperature of 25 ± 0.5 °C, and all pots received the same amount of deionized water. The experiment was carried out in November 2019.

Destructive samplings were sampled at 7 and 15 days after N fertilization. Shoots were firstly collected from each pot at destructive harvesting, then put at 105 °C for 30 min to terminate any metabolic activity, dried at 65 °C for 24 h, weighed and analyzed for C and N concentration. Roots in soils were carefully sampled from soil by hands at destructive harvesting, then washed out with tap water and then rinsed 3 times with deionized water. The roots were scanned on an Epson root scanner (Epson Perfection V700 photo, Nagano, Japan), then analyzed by the WIN-RHIZO software (Regent Instruments Inc., Quebec, Canada), and finally dried at 65 °C until constant weight for the dry weight an C and N determination. A CN Elemental Analyzer (Element High TOC II, Berlin, Germany) was used to determine the TC and TN contents in shoot and root.

Soil in each pot was fully sampled, thoroughly homogenized and kept in 4 °C for the analysis of NH$_4^+$-N, NO$_3^-$-N and Glu content. NH$_4^+$-N and NO$_3^-$-N in soil were extracted with a 2 M KCl solution (fresh soil:extracts = 1:5) and measured using a flow injection system (SEAL AutoAnalyzer 3, Norderstedt, Germany). The determination of Glu content was carried out by High-Performance Liquid Chromatography (VARIAN, Palo Alto, CA, USA) using AquaSep C85μ chromatogram column; samples were eluted with 0.05 mol/L KH$_2$PO$_4$ solution (pH = 2.4) at 0.6 mL/min and detected at 210 nm, and the chromatographically pure Glu was used as a standard.

Differences in root traits and soil variables between the two treatments were compared using one-way analysis of variance (ANOVA). Spearman correlation analysis was used to examine the relationship between the root morphological traits and soil variables. Two-way ANOVAs were used to analyze the effects of γ-PGA, N fertilizer types and their interactions on root traits and soil: NH$_4^+$-N, NO$_3^-$-N, NH$_4^+$-N/NO$_3^-$-N and Glu. The multiple comparisons were conducted by least significant difference (LSD) test. All statistical analyses were carried out in SPSS 19.0 (SPSS Inc., Chicago, IL, USA) at $p < 0.05$.

## 3. Results

### 3.1. Root Dry Weight and the Ratio of Root to Shoot under the Effect of γ-PGA

γ-PGA had a significant promotion effect on the root dry weight whether it was applied with CO(NH$_2$)$_2$ or (NH$_4$)$_2$SO$_4$ at both Day 7 and Day 15 (Figure 1A). The ANOVA indicated that N fertilizer forms also had a significant effect on root dry weight at Day 15 (Figure 1A). There was a significant difference in root dry weight of each treatment between Day 7 and Day 15 (Figure 1A). No significant difference in the ratio of root to shoot was observed among treatments both at Day 7 and Day 15 (Figure 1B).

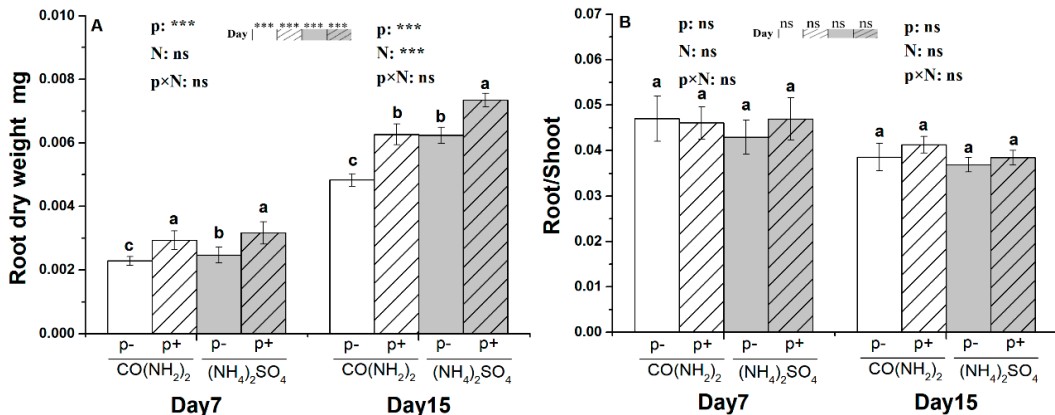

**Figure 1.** Effects of poly γ-glutamic acid (γ-PGA) and nitrogen (N) fertilizer forms on the root dry weight and the ratio of root to shoot (root/shoot). (**A**) Root dry weight, (**B**) Root/Shoot. Error bars represent standard errors (SE, *n* = 10). Results are presented as mean ± SE; p: γ-PGA; N: nitrogen fertilizer forms; p−: without γ-PGA addition; p+: with γ-PGA addition; day: sampling day; ***: $p \leq 0.001$; ns: not significant. Different lowercase letters represent the significant difference among the four treatments.

*3.2. Root Total Carbon Content, Total Nitrogen Content and the Ratio of Carbon to Nitrogen under the Effect of γ-PGA*

γ-PGA significantly increased the TC content of roots at Day 7 and Day 15 when applied with $CO(NH_2)_2$. Both the γ-PGA and the interactions between γ-PGA and N fertilizer forms had significant effects on the TC content of roots at Day 7 (Figure 2A), while γ-PGA significantly decreased the TC content of roots when applied with $(NH_4)_2SO_4$ at Day 7 (Figure 2A). In addition, γ-PGA significantly decreased the TN content of roots at Day 7 and Day 15 regardless of being applied with $CO(NH_2)_2$ or $(NH_4)_2SO_4$ (Figure 2B), yet the total amount of N in roots was significantly enhanced by γ-PGA (data not shown). The γ-PGA, the N fertilizer forms and the interactions between γ-PGA and N fertilizer forms all had significant effects on the TN content of roots both at Day 7 and Day 15 (Figure 2B). Additionally, the ratio of carbon to nitrogen (C/N) in roots was significantly improved by γ-PGA at both Day 7 and Day 15, and also significantly affected by the N fertilizer forms and the interactions between γ-PGA and N fertilizer forms at Day 7 (Figure 2C). Sampling time exerted significant influences on the root TC, TN and C/N of each treatment (Figure 2).

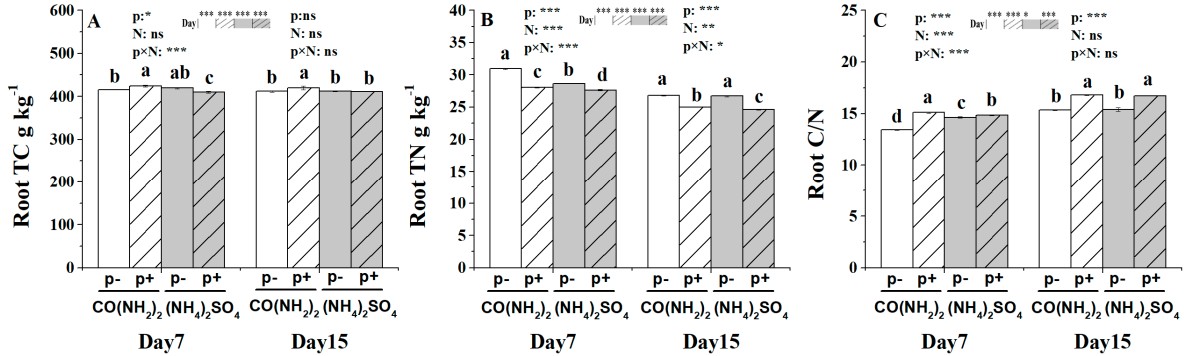

**Figure 2.** Effects of γ-PGA and N fertilizer forms on the total carbon (TC) content, the total nitrogen (TN) content and the ratio of C to N (C/N) in roots. (**A**) Root TC, (**B**) Root TN, (**C**) Root C/N. Error bars represent standard errors (SE, *n* = 10). Results are presented as mean ± SE; p: γ-PGA; N: nitrogen fertilizer forms; p−−: without γ-PGA addition; p+: with γ-PGA addition; day: sampling day; *: $p \leq 0.05$; **: $p \leq 0.01$; ***: $p \leq 0.001$; ns: not significant. Different lowercase letters represent the significant difference among the four treatments.

### 3.3. Root Morphological Traits under the Effect of γ-PGA

Root length was increased in the treatment with γ-PGA and $CO(NH_2)_2$ application compared to with $CO(NH_2)_2$ application, but only significant at Day 15 (Figure 3A). However, in comparison with treatment with $(NH_4)_2SO_4$ application, root length was significantly decreased in the treatment with γ-PGA and $(NH_4)_2SO_4$ application at Day 7 (Figure 3A). At Day 15, the treatment with γ-PGA and $CO(NH_2)_2$ application had significantly increased the total surface area relative to the treatment with $CO(NH_2)_2$ application (Figure 3C). Specific root length and specific root surface area both were greater, but not significant, in the treatment with γ-PGA and $CO(NH_2)_2$ application than with $CO(NH_2)_2$ application at Day 7 and Day 15, while they were significantly smaller in the treatment with γ-PGA and $(NH_4)_2SO_4$ application than with $(NH_4)_2SO_4$ application at Day 7 (Figure 3B,D). Root average diameter was reduced by γ-PGA, regardless of whether it was applied with $CO(NH_2)_2$ or $(NH_4)_2SO_4$; however, it was only significantly reduced at Day 15 between treatments with $CO(NH_2)_2$ fertilization (Figure 3E). At Day 7, root volume was significantly reduced in the treatment with γ-PGA and $(NH_4)_2SO_4$ application compared to with $(NH_4)_2SO_4$ application (Figure 3F). More root tips and forks were found in the treatment with γ-PGA and $CO(NH_2)_2$ application than with $CO(NH_2)_2$ application, while only significant at Day 15. Conversely, less root tips and forks were found in treatment with γ-PGA and $(NH_4)_2SO_4$ application than with $(NH_4)_2SO_4$ application, while only significant at Day 7. Two-way ANOVA results showed that root length, total surface area and diameter were all significantly affected by γ-PGA addition at Day 7 and Day 15, and root length, total surface area, diameter and forks were all significantly affected by N fertilizer forms at Day 15 (Figure 3A,C,E,H). Sampling time exerted significant influences on the root length, surface area, volume, tips and forks of the four treatments (Figure 3A,C,F,G,H).

### 3.4. Soil Available N Content and Glutamic Acid Content under the Effect of γ-PGA

Soil $NH_4^+$-N content in the treatment with γ-PGA and $CO(NH_2)_2$ application relative to that with $CO(NH_2)_2$ application was significantly reduced at Day 7, but subsequently significantly increased at Day 15. In addition, soil $NH_4^+$-N content in the treatment with γ-PGA and $(NH_4)_2SO_4$ application relative to that with $(NH_4)_2SO_4$ application was significantly reduced both at Day 7 and at Day 15 (Figure 4A). Soil $NO_3^-$-N content was significantly reduced in the treatment with γ-PGA and $CO(NH_2)_2$ application compared to with $CO(NH_2)_2$ application both at Day 7 and at Day 15, and significantly reduced in the treatment with γ-PGA and $(NH_4)_2SO_4$ application compared to with $(NH_4)_2SO_4$ application at Day 15 (Figure 4B). The ratio of $NH_4^+$-N to $NO_3^-$-N ($NH_4^+$-N/$NO_3^-$-N) was significantly higher in the treatment with γ-PGA and $(NH_4)_2SO_4$ application compared to the treatment with $(NH_4)_2SO_4$ application at Day 15 (Figure 4C). In addition, the $NH_4^+$-N/$NO_3^-$-N in treatments with $CO(NH_2)_2$ fertilization, regardless of whether γ-PGA was applied or not, were significantly lower than those with $(NH_4)_2SO_4$ fertilization both at Day 7 and Day 15 (Figure 4C). The Glu content in soil was significantly improved by γ-PGA addition at Day 7 and Day 15, no matter whether γ-PGA was applied with $CO(NH_2)_2$ or $(NH_4)_2SO_4$ (Figure 4D). γ-PGA, N fertilizer forms and sampling time all significantly affected the $NH_4^+$-N and $NO_3^-$-N content in soil (Figure 4A,B), and both N fertilizer forms and sampling time had significant effects on the $NH_4^+$-N/$NO_3^-$-N in soil (Figure 4C). γ-PGA had a significant effect on Glu content in soil, and sampling time only exerted a significant influence on the Glu content in soil with Glu addition (Figure 4D).

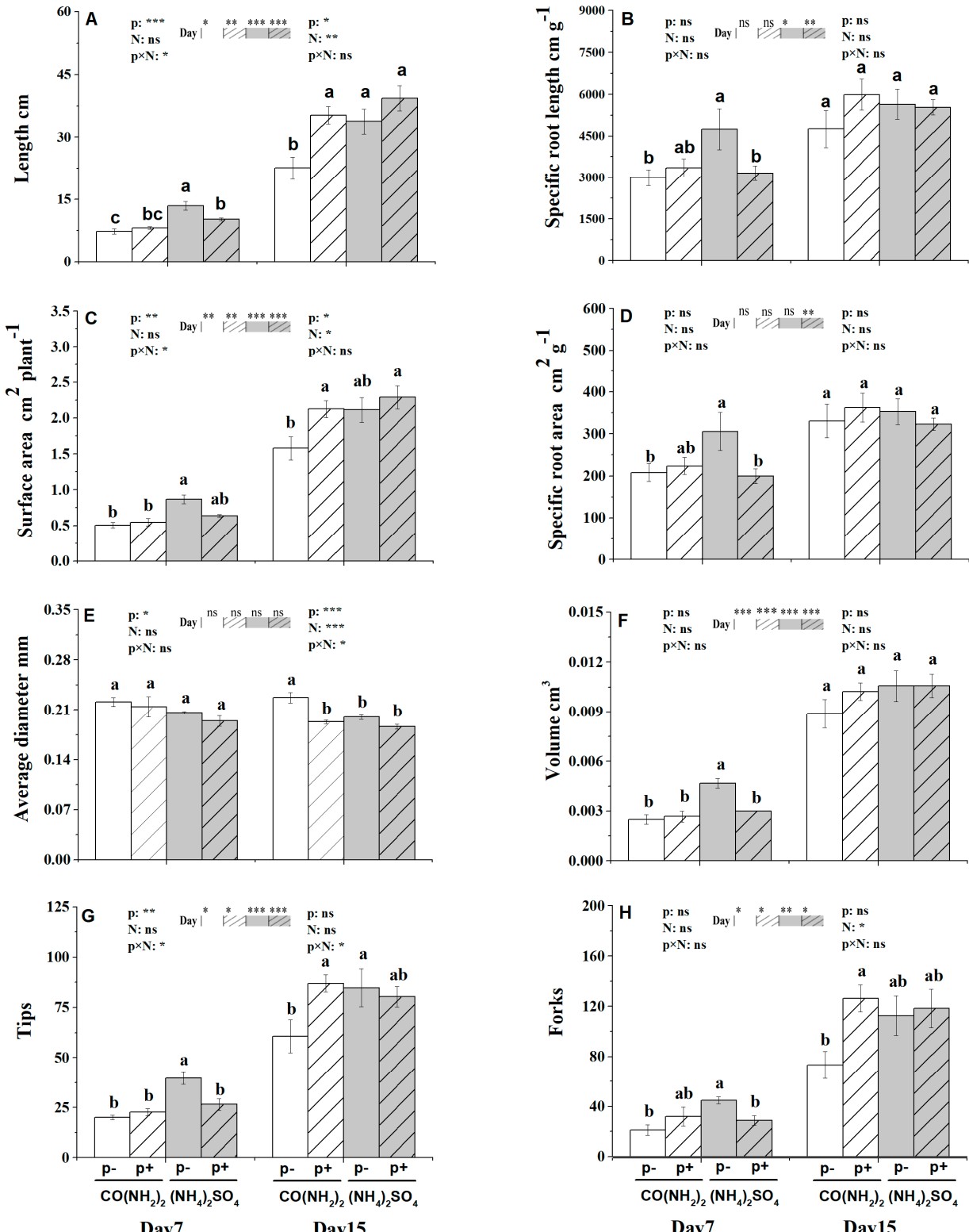

**Figure 3.** Effects of γ-PGA and N fertilizer forms on the root morphological traits. (**A**) Length, (**B**) Specific root length, (**C**) Surface area, (**D**) Specific root area, (**E**) Average diameter, (**F**) Volume, (**G**) Tips, (**H**) Forks. Error bars represent standard errors (SE, *n* = 10). Results are presented as mean ± SE; p: γ-PGA; N: nitrogen fertilizer forms; p−: without γ-PGA addition; p+: with γ-PGA addition; day: sampling day; *: $p \leq 0.05$; **: $p \leq 0.01$; ***: $p \leq 0.001$; ns: not significant. Different lowercase letters represent the significant difference among the four treatments.

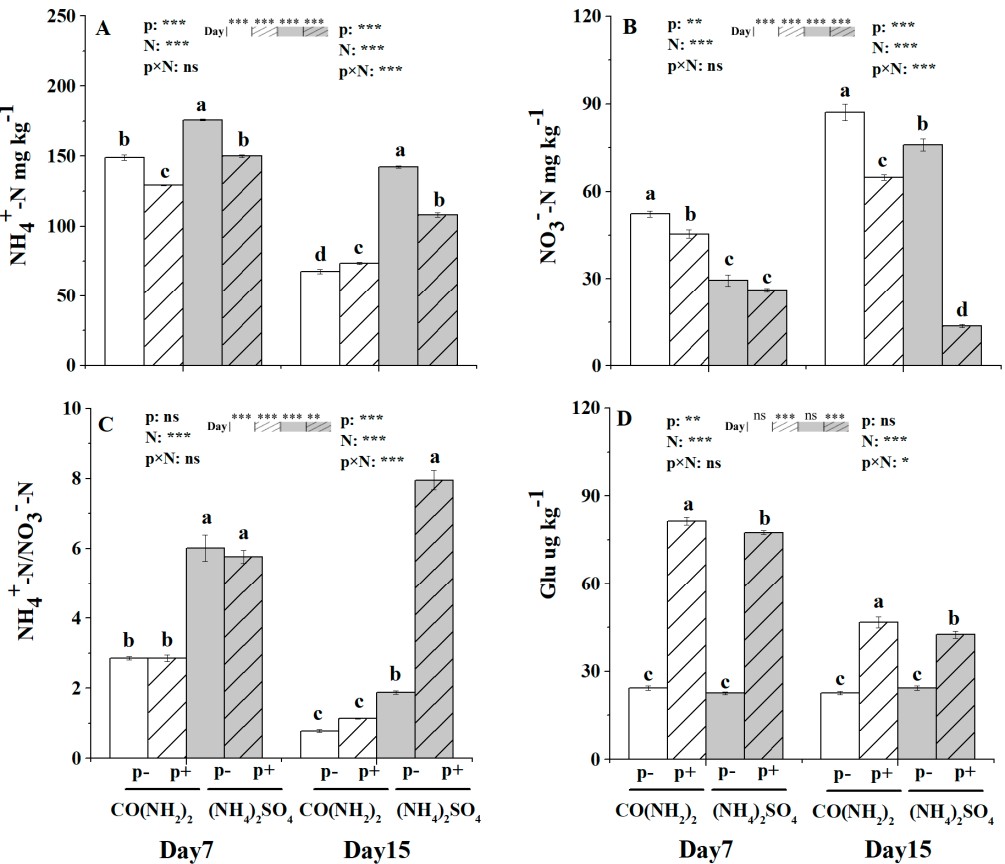

**Figure 4.** Effects of γ-PGA and N fertilizer forms on the content of ammonium ($NH_4^+$-N), nitrate ($NO_3^-$-N) and glutamic acid (Glu) and the ratio of $NH_4^+$-N/$NO_3^-$-N ($NH_4^+$-N/$NO_3^-$-N) in soil. (**A**) $NH_4^+$-N, (**B**) $NO_3^-$-N, (**C**) $NH_4^+$-N/$NO_3^-$-N, (**D**) Glu. Error bars represent standard errors (SE, *n* = 10). Results are presented as mean ± SE; p: γ-PGA; N: nitrogen fertilizer forms; p−: without γ-PGA addition; p+: with γ-PGA addition; day: sampling day; *: $p \leq 0.05$; **: $p \leq 0.01$; ***: $p \leq 0.001$; ns: not significant. Different lowercase letters represent the significant difference among the four treatments.

*3.5. Relationships between Root Traits and Soil Variables*

Spearman analysis results showed that root dry weight was positively correlated with the C/N in root and all the root morphological traits except for the negative correlation with root diameter, and negatively correlated with the root TN content, root TC content and soil $NH_4^+$-N content (Table 1). A significant negative correlation was observed between the soil $NH_4^+$-N content and the soil $NO_3^-$-N content, the C/N in roots and all the root morphological traits except root diameter (Table 1). In addition, a significant positive correlation was observed between the soil $NH_4^+$-N content and the TC content of roots and the $NH_4^+$-N/$NO_3^-$-N in soil (Table 1). Soil $NO_3^-$-N content had a significant negative correlation with the soil $NH_4^+$-N content and the $NH_4^+$-N/$NO_3^-$-N and had a significant positive correlation with the root volume and the root tips (Table 1). The $NH_4^+$-N/$NO_3^-$-N in soil had a significant negative correlation with the root volume and the root tips (Table 1). There was no significant correlation between the Glu content in soil and all the root morphological traits and soil variables (Table 1), but Glu content had a significant and positive correlation relationship with root dry weight, the root C/N, the root length, the root surface area and the root tips at Day 7, and had a significant and negative correlation with the root TC content and the soil $NH_4^+$-N content at Day 7 (data not shown). In addition, there was a significant positive correlation between any two of the root morphological traits, except for the root diameter, which had a negative correlation with the other root morphological traits (Table 1).

**Table 1.** Correlations between root traits and soil variables under the effect of γ-PGA and N fertilizers.

| | RDW | RS | TN | TC | CN | NH$_4^+$-N | NO$_3^-$-N | NH$_4^+$-N/NO$_3^-$-N | Glu | RL | SRL | RA | SRA | Diameter | Volume | Tip | Fork |
|---|---|---|---|---|---|---|---|---|---|---|---|---|---|---|---|---|---|
| RDW | 1 | | | | | | | | | | | | | | | | |
| RS | −0.38 | 1 | | | | | | | | | | | | | | | |
| TN | −0.440 * | 0.14 | 1 | | | | | | | | | | | | | | |
| TC | −0.889 ** | 0.502 * | 0.342 | 1 | | | | | | | | | | | | | |
| C/N | 0.807 ** | −0.491 * | −0.019 | −0.919 ** | 1 | | | | | | | | | | | | |
| NH$_4^+$-N | −0.650 ** | 0.234 | 0.086 | 0.695 ** | −0.807 ** | 1 | | | | | | | | | | | |
| NO$_3^-$-N | 0.095 | −0.155 | 0.038 | −0.044 | 0.182 | −0.435 * | 1 | | | | | | | | | | |
| NH$_4^+$-N/NO$_3^-$-N | −0.095 | 0.131 | −0.126 | 0.122 | −0.295 | 0.549 * | −0.961 ** | 1 | | | | | | | | | |
| Glu | 0.056 | 0.202 | 0.178 | −0.163 | 0.206 | −0.097 | −0.38 | 0.217 | 1 | | | | | | | | |
| RL | 0.911 ** | −0.560 * | −0.311 | −0.897 ** | 0.876 ** | −0.694 ** | 0.17 | −0.197 | 0.122 | 1 | | | | | | | |
| SRL | 0.581 * | −0.489 * | −0.07 | −0.651 ** | 0.705 ** | −0.550 * | 0.23 | −0.287 | 0.079 | 0.767 ** | 1 | | | | | | |
| RA | 0.889 ** | −0.507 * | −0.307 | −0.878 ** | 0.854 ** | −0.724 ** | 0.222 | −0.246 | 0.067 | 0.970 ** | 0.815 ** | 1 | | | | | |
| SRA | 0.561 * | −0.379 | 0.006 | −0.609 * | 0.693 ** | −0.613 ** | 0.328 | −0.376 | 0.058 | 0.737 ** | 0.957 ** | 0.811 ** | 1 | | | | |
| Diameter | −0.412 * | 0.363 | 0.214 | 0.459 * | −0.368 | −0.05 | 0.322 | −0.286 | −0.281 | −0.465 * | −0.289 | −0.311 | −0.08 | 1 | | | |
| Volume | 0.735 ** | −0.497 * | −0.073 | −0.781 ** | 0.855 ** | −0.794 ** | 0.415 * | −0.464 * | 0.037 | 0.855 ** | 0.794 ** | 0.855 ** | 0.830 ** | −0.192 | 1 | | |
| Tips | 0.818 ** | −0.439 * | −0.147 | −0.830 ** | 0.883 ** | −0.776 ** | 0.419 * | −0.466 * | 0.046 | 0.877 ** | 0.730 ** | 0.903 ** | 0.755 ** | −0.222 | 0.855 ** | 1 | |
| Forks | 0.858 ** | −0.581 * | −0.144 | −0.836 ** | 0.864 ** | −0.654 ** | 0.269 | −0.272 | 0.031 | 0.938 ** | 0.768 ** | 0.948 ** | 0.776 ** | −0.304 | 0.849 ** | 0.913 ** | 1 |

RDW, RS, TN, TC, C/N, NH$_4^+$-N, NO$_3^-$-N, NH$_4^+$-N/NO$_3^-$-N, Glu, RL, SRL, RA and SRA represent root dry weight, the ratio of root to shoot, total nitrogen content in root, total carbon content in root, the ratio of carbon to nitrogen in root, soil ammonium content, soil nitrate content, the ratio of ammonium to nitrate in soil, glutamic acid content in soil, root length, specific root length, root surface area and specific surface area, respectively. ** Correlation is significant at the 0.01 level; * correlation is significant at the 0.05 level.

## 4. Discussion

γ-PGA significantly increased the root dry weight of Pakchoi seedlings (*Brassica rapa* subsp. *chinensis*) when applied with N fertilizers, which was consistent with the previous results [20–23], while γ-PGA had a greater promotion on root dry weight when applied with $CO(NH_2)_2$ compared to $(NH_4)_2SO_4$, similar to the result reported by Zhang et al. (2022) [19]. Root dry weight was positively correlated with all root morphological traits tested in this study, except for the root diameter, which was identical to the previous result [25]. This result demonstrated that the root growth was in pace with the change of root morphological traits. A higher C/N and a lower TN content were observed in roots with γ-PGA addition, and this may suggest that γ-PGA contributes more aboveground C metabolite allocation to roots compared with the N metabolite, yet the N uptake amount by roots was still higher in roots with γ-PGA addition (data not shown), as also confirmed in our previous studies [19,20]. The role of Glu in plant C metabolism cannot be overstated [26], and Zhang et al. (2017) observed that Glu decomposed from γ-PGA can be directly absorbed by plant roots and that the increase in Glu content in the external and internal areas of roots may accelerate the C investment to roots and therefore provide a possible reason for the higher C:N in roots.

Plasticity of the root morphological traits is critical for N acquisition in plant and plant growth. γ-PGA significantly increased the total root length, root surface area, tips and forks at Day 15 when applied with $CO(NH_2)_2$. However, surprisingly, γ-PGA significantly suppressed all morphology traits (except diameter) at Day 7 when applied with $(NH_4)_2SO_4$. Root total length and root surface area represent the explorative volume of roots in soil, which is one of the important factors that determine the nutrient acquisition ability of roots [27–29], and the tips and forks represent the production of new roots and root branching. The improved total root length, surface area, tips and forks suggested that the proliferation of roots and also the nutrient acquisition ability of roots were enhanced by γ-PGA when combined with $CO(NH_2)_2$ fertilization. In addition, the suppressed root traits indicated that the proliferation of roots was hampered by γ-PGA when combined with $(NH_4)_2SO_4$ fertilization. Different from other root traits, root diameter became slenderer after the addition of γ-PGA with N fertilizer, but only significant at Day 15 when applied with $CO(NH_2)_2$. This result coincided with higher root biomass and outgrowth in the treatment with the combined application of γ-PGA and $CO(NH_2)_2$, because slimmer root diameter meant the increase in production of fine roots [30] and resulted in a thinner root system with a larger surface area, thus conferring greater nutrient uptake per unit of root mass [28]. Additionally, the above results suggest that γ-PGA stimulated root elongation and outgrowth at a relative late pace when applied with $CO(NH_2)_2$ and inhibited root growth at a relative early pace when applied with $(NH_4)_2SO_4$.

The promotional effect of γ-PGA on root growth when applied with $CO(NH_2)_2$ could be comprehensively explained by the following facts. Firstly, L-Glu can act as an external signal to inhibit primary root growth and stimulate the lateral root growth, and even large excess concentrations of L-Glu produce neither severe growth abnormalities nor toxicity symptoms to roots [15]. The sharp increase in Glu content in soil (Figure 4D) after γ-PGA addition may be able to stimulate the outgrowth of lateral roots, and thus resulted in the increase in total root length. As a kind of dicotyledon plant, the root system of Pakchoi seedlings (*Brassica rapa* subsp. *chinensis*) is composed of one primary root and many lateral roots [31], therefore the inhibition of Glu on the primary root can be negligible. In this study, the soil Glu content showed no significant correlation with soil properties and root traits (Table 1), but the soil Glu content at Day 7 was positively related with the dry weight, C/N, length, surface area and tips of roots and was significantly negatively related with the root TC content and the soil $NH_4^+$-N content (data not shown). These results indicated that Glu may stimulate root growth mainly at the early stage after its addition. Apart from Glu, soil mineral N status ($NH_4^+$-N and $NO_3^-$-N) modulated root growth notably, as the excess mineral N in soil restrained plant root growth and the mild deficiency of mineral N in soil stimulated root growth [6,7]. The $NH_4^+$-N content in soil with $CO(NH_2)_2$ fertilization

was significantly reduced after the addition of $\gamma$-PGA, and recent studies suggested that external $CO(NH_2)_2$ might serve as an effective signal molecule for ameliorating $NH_4^+$-N suppressed root growth [32,33]. Therefore, the combined application of $\gamma$-PGA and $CO(NH_2)_2$ is likely to be conducive to preventing the root from the inhibition derived from high $NH_4^+$-N content. What is more, the $NO_3^-$-N content in soil with $CO(NH_2)_2$ fertilization was also significantly reduced after the addition of $\gamma$-PGA, and the moderate or mild deficiency of $NO_3^-$-N content in soil was beneficial to the development of lateral root branching and density [6,7]. In addition, by antagonizing the inhibitory effect of Glu, $NO_3^-$-N was also reported to stimulate primary root growth indirectly [16], and thus would relieve the inhibition of Glu on primary root length. Above all, the promoted root biomass and morphology by $\gamma$-PGA may comprehensively result from the lowered $NH_4^+$-N and $NO_3^-$-N content, the greater L-Glu content and the $CO(NH_2)_2$ in soil after its addition with $CO(NH_2)_2$.

The inhibition on root growth by $\gamma$-PGA addition when applied with $(NH_4)_2SO_4$ may be mainly caused by the inhibition function of the high $NH_4^+$ content in soil. Peng et al. (2012) observed that a high base N fertilizer application (175 kg ha$^{-1}$), accompanied by a high level of mineral N content in soil, inhibited maize root growth in the early growth period (45 days after fertilization) [34]. $NH_4^+$-N content was significantly higher in soils with $(NH_4)_2SO_4$ fertilization (AS and ASP treatment) than with $CO(NH_2)_2$ fertilization (U and UP treatment), and although $NH_4^+$-N content in soil was significantly lowered after the addition of $\gamma$-PGA, it was still much larger than $5-10$ mM to inhibit root elongation [8,10]. Pearson correlation analysis further showed that $NH_4^+$-N was negatively correlated with root dry weight and all morphology traits, which was consistent with the previous studies [35–37]: those earlier studies all presented a significantly negative correlation between soil $NH_4^+$-N content and root length. In addition, a previous research study found that Glu addition can intensify the inhibition function of high $NH_4^+$-N stress on the root growth [11], which may further inhibit root growth.

The contradictory results between the inhibition effect of $\gamma$-PGA on root morphological traits and the promotion effect of $\gamma$-PGA on root biomass and N content when applied with $(NH_4)_2SO_4$ remind us of the possibility that there would be other strategies employed by roots to increase nutrient acquisition, such as mycorrhizal fungi or root exudation [38–40]. Therefore, it is worthy to conduct a further study to systematically reveal the function and mechanism of $\gamma$-PGA, as a N fertilizer synergist, on root nutrient acquisition.

## 5. Conclusions

It is obviously demonstrated in this study that $\gamma$-PGA significantly increased the root dry weight and C/N, and significantly reduced the root TN content regardless of whether it was applied with $CO(NH_2)_2$ or $(NH_4)_2SO_4$. $\gamma$-PGA stimulated root proliferation when applied with $CO(NH_2)_2$ yet inhibited it when applied with $(NH_4)_2SO_4$. The paradoxical facts between the increase in root biomass and N acquisition and the inhibition in root morphology demonstrated the existence of other N foraging strategies when $\gamma$-PGA was applied with $(NH_4)_2SO_4$. This opposite response of root morphology on $\gamma$-PGA when applied with different N fertilizer reminds us that the N foraging strategies change variably dependent on soil available N ($NO_3^-$-N, $NH_4^+$-N, Glu) status, and the promotion pathways of $\gamma$-PGA on foraging N nutrients are complex and remain to be elucidated comprehensively in future study.

**Author Contributions:** Conceptualization, L.Z. (Lei Zhang) and L.Z. (Lili Zhang); methodology, X.D. and Y.S (Yu Sun).; validation, L.Z. (Lili Zhang), Y.S. (Yu Sun) and L.Z. (Lei Zhang); formal analysis, L.W.; data curation, Z.W. and L.W.; writing—original draft preparation, L.Z. (Lei Zhang); writing—review and editing, L.W. and Z.W.; supervision, L.Z. (Lili Zhang) and Y.S. (Yuanliang Shi); project administration, L.Z. (Lili Zhang) and Y.S. (Yuanliang Shi); funding acquisition, L.Z. (Lei Zhang), L.W. and L.Z (Lili Zhang). All authors have read and agreed to the published version of the manuscript.

**Funding:** This research was funded by the National Natural Science Foundation of China, grant number 41807090, the Shenyang Young and Middle-aged Science and Technology Innovation Talents Support Program Project, grant number RC220124, the Strategic Priority Research Program of the Chinese Academy of Sciences, grant numbers XDA28090200 and XDA28100200, and the K.C. Wong Education Foundation—Research and application of environmental-friendly polymer compound fertilizers, grant number 2022-5.

**Data Availability Statement:** The data presented in this study are available on request from the author.

**Acknowledgments:** Experimental soils offered by J.K. Wang from Shenyang Agricultural University are much appreciated.

**Conflicts of Interest:** The authors declare no conflict of interest.

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
