# Peer review of "Opposite Root Morphological Responses of Chinese Cabbage to Poly-γ-glutamic Acid When Applied with Urea and Ammonium Sulphate"

_agronomy, doi:10.3390/agronomy13020608_

Round 1

Reviewer 1 Report

The manuscript agronomy-2175691 is a nice piece of work, well written and easy to understand. An experiment was conducted to investigate the response of root morphmological traits to γ-polyglutamic acid when applied in combination with urea or ammonium sulfate. The result is of agronomic value because polyglutamic acid promoted root growth and stimulated root proliferation when applied with urea, whilst it inhibited root growth when applied with ammonium sulfate. A useful discussion of the phenomenon is provided. The work is of agronomic significance.

However, some minor improvements are needed:

L.19, please delete “and”; it is “or (NH4)2SO4”.

L.25, the same. Also, “, this provides” -> “. This finding provides”.

L.34, Root is responsible

L.35, which is an adaptive

L.39, L.47, Guo et al., 2017)). Please delete one bracket.

L.56, please write it as 0.050 mM, because in L.59 you use mM.

L.66, please correct “α-amoin and the γ-carbonly”

L.84, “we conducted”

L.86, please replace and with or: “or ammonium”

L.100, please provide the manufacturer of the PGA.

L.110, instead of “killed” -> put at 105 oC to terminate any metabolic activity

L.116, “, and C and N determination” (not measurement)

L.136, in Fig.1 it is pxN, not PxN

L.218, delete the 2. Materials …

L.248, please replace “And” with “Also,”

Reviewer 2 Report

This is the revision of the manuscript number agronomy-2175691 Title: “ Opposite root morphological responses to poly-γ-glutamic acid when apply it with urea and ammonium sulphate”, proposed by MsAnita Pizurica and colleagues for consideration for publication in Agronomy

The manuscript raises an interesting topic regarding the evaluation of γ-PGA with respect to different types of conventional N fertilizers in the influence of root growth of a Brassica species, even so, I would like certain notes to be taken for the publication of the manuscript manuscript.

Introduction:

Line 39 and 52 According to the citation rules of the agronomy journal, I think that the one shown in the authors Gruber and Walch-Liu is not the correct form.

Results and discussion:

Graphics It would be interesting if the data shown in the graphs indicated statistics (ANOVA) between the values of the destructive taps on day 7 and the values shown corresponding to day 15.

Line 213-229 There is an error in point 3.5 I think it does not correspond to the manuscript and should be eliminated.

Due to the importance that N shows with respect to the enzymatic activity present in the soil, it would have been important to increase the parameters accounted for with the analysis of urease activity in soil samples.

General comments:

The author refers to the interaction of two N fertilizers to γ-PGA, these studies could be carried out in uncontrolled environmental conditions, such as carrying out the study in the field, because the effect of temperature could change the dynamics of N on the floor.

Reviewer 3 Report

It is interenting article that anlaize the efect of γ-PGA in the root morpholical response by winrhyzo. 

In general, γ-PGA shows promising application potential in agricultural use for its anionic, biodegradable and biocompatible properties. Howeber, the studies es very short time to know the agronomist effect. It would be intested to analize the increase in the production for exampleand no just the responce in the vegetative growth stage. 

I suggest to sent the article a plant nutrition journal or similar. 

Reviewer 4 Report

You need to make a deep analysis to differentiate your findings from the previous studies special yours. Show the readers the differences between this study and your previous study. Do that in the introduction, and show the gap of the study.

Round 2

Reviewer 3 Report

The article can be accepted at the present form

Reviewer 4 Report

The paper now in a good shape.